# Does a Vegetarian Diet Affect the Levels of Myokine and Adipokine in Prepubertal Children?

**DOI:** 10.3390/jcm10173995

**Published:** 2021-09-03

**Authors:** Jadwiga Ambroszkiewicz, Joanna Gajewska, Joanna Mazur, Witold Klemarczyk, Grażyna Rowicka, Mariusz Ołtarzewski, Małgorzata Strucińska, Magdalena Chełchowska

**Affiliations:** 1Department of Screening Tests and Metabolic Diagnostics, Institute of Mother and Child, 01-211 Warsaw, Poland; joanna.gajewska@imid.med.pl (J.G.); mariusz.oltarzewski@imid.med.pl (M.O.); magdalena.chelchowska@imid.med.pl (M.C.); 2Department of Humanization in Medicine and Sexology, Collegium Medicum, University of Zielona Gora, 65-726 Zielona Gora, Poland; j.mazur@cm.uz.zgora.pl; 3Department of Nutrition, Institute of Mother and Child, 01-211 Warsaw, Poland; witold.klemarczyk@imid.med.pl (W.K.); grazyna.rowicka@imid.med.pl (G.R.); malgorzata.strucinska@imid.med.pl (M.S.)

**Keywords:** myostatin, irisin, omentin, visfatin, lacto-ovo-vegetarian diet

## Abstract

Myokines are cytokines secreted by muscle and exert autocrine, paracrine, or endocrine effects. Myokines mediate communication between muscle and other organs, including adipose tissue. The aim of the study was to assess serum myokines and their relationships with adipokines and anthropometric and nutritional parameters in children following vegetarian and omnivorous diets. One hundred and five prepubertal children were examined. Among them there were 55 children on a vegetarian diet and 50 children on an omnivorous diet. Concentrations of myokines (myostatin, irisin) and adipokines (leptin, adiponectin, omentin, visfatin) in serum were determined by enzyme-linked immunosorbent assay (ELISA). We observed comparable median values of serum myokines and adipokines (except of leptin concentration) in both of the studied groups of children. We also found several correlations between myokine and adipokine levels and certain nutritional parameters. Serum myostatin was positively correlated with omentin levels in vegetarians and omnivores (*p* = 0.002). Serum irisin was positively associated with omentin (*p* = 0.045) levels in omnivores and inversely with visfatin concentration (*p* = 0.037) in vegetarians. Myostatin concentration was negatively correlated with the percentage of energy from protein (*p* = 0.014), calcium (*p* = 0.046), and vitamin A (*p* = 0.028) intakes in vegetarians and with dietary vitamin C (*p* = 0.041) and vitamin E (*p* = 0.021) intakes in omnivores. In multivariate regression analyses, positive correlations of serum myostatin with omentin levels were revealed in both study groups (β = 0.437, *p* < 0.001 for vegetarians; and β = 0.359, *p* = 0.001 for omnivores). Consuming a lacto-ovo-vegetarian diet did not influence serum levels of myokines (myostatin, irisin) and adipokines such as adiponectin, visfatin, and omentin in prepubertal children. However, leptin levels were significantly lower in vegetarians compared with omnivores. The observed significant positive correlations between myostatin and omentin concentrations might suggest tissue cross-talk between skeletal muscle and fat tissue. Further studies, carried out in a larger group of children following different dietary patterns, could be important to fully understand the relations between muscle, adipose tissues, and nutrition.

## 1. Introduction

A proper diet is one of the important factors that determine age-appropriate body weight and body composition, including fat, lean, and bone mass, which is why children’s diets should be safe and prevent nutrient deficiencies [1,2]. Recently, many vegetarian parents want their children to adopt their diet [3,4,5]. The lacto-ovo-vegetarian diet, which includes milk, milk products, and eggs is the most common variant. Generally, a well-balanced plant-based diet with an adequate intake of high-quality protein, calcium, and vitamins (especially B_12_ and D) can satisfy all the nutritional needs of a growing child [6,7].

Both muscle and fat tissues are endocrine organs that secrete many biologically active factors, known as myokines and adipokines, implicated in the regulation of energy homeostasis and metabolism [8,9]. Myokines are cytokines produced by muscle fibers (myocytes), mainly in response to physical activity. Among them, myostatin and irisin appear to have promising clinical value. Myostatin, a protein belonging to the family of transforming β growth factor (TGF-β), plays a role in the regulation of skeletal muscle mass through myoblast proliferation and differentiation [10,11,12,13]. Another myokine, irisin, is a peptide (112–123 amino acids, 22 kDa) cleaved from the extracellular domain of Fndc5 (fibronectin type III domain containing proteins 5) by stimulation with PGC-1α (peroxisome proliferator-activated receptor gamma coactivator 1-α), expressed in skeletal muscle and other tissues [14,15]. Recent experimental and observational studies have indicated that irisin is a myokine that induces white adipose tissue browning, energy expenditure, improved glucose tolerance, and enhanced thermogenesis [16,17]. Besides its hormonal function, it may act as a transmembrane signaling protein that mediates the cross-talk of skeletal muscle to other tissues; however, this hypothesis remains to be confirmed [18]. It is important to note that irisin is not a protein exclusively secreted by muscle tissue. It is also secreted by white adipose tissue; thus, irisin is not only a myokine but also an adipokine.

Apart from the well-known adipokines, such as leptin and adiponectin, fat tissue also releases other adipokines, including visfatin and omentin. Visfatin, expressed mainly by visceral adipose tissue, regulates adipocyte differentiation and is involved in the control of weight [19]. Serum visfatin concentrations depend on lifestyle interventions, including exercise and diet. As a multifunctional protein, visfatin may act as a hormone, cytokine, and enzyme-nicotinamide phosphoribosyltransferase (Nampt) [20]. Omentin is an adipokine that has anti-inflammatory properties. It plays a role in adipocyte differentiation and maturation, metabolism and immune response regulation, inflammation, and insulin resistance [21]. Its decreased level was observed in obese compared with lean subjects, suggesting that omentin concentration may be predictive of the metabolic consequences associated with obesity [22].

The relationships between muscle and adipose tissues are very important [23,24], especially in children who are in the developmental period. Many studies have shown the beneficial effects of a plant-based diet on decreased risk of several diseases, but the underlying mechanisms for this influence are not fully understood [25,26]. No data exist concerning myokines levels and their relations with the adipokine profile in vegetarians.

The aim of this study was to assess serum concentrations of myokines (myostatin, irisin) and to analyze their relationships with some adipokine levels and anthropometric and nutritional parameters in children with different eating styles: vegetarians and omnivores.

## 2. Materials and Methods

### 2.1. Subjects

We examined 105 prepubertal healthy children. All studied children were Caucasians. Among them, there were 55 (52.7% male, 47.3% female, age 5–9 years) children on a vegetarian diet. The inclusion criteria were: being in the prepubertal period, on a lacto-ovo-vegetarian diet from birth, apparently healthy without disorders in terms of development and nutrition. The exclusion criteria were: low birth weight, gastrointestinal diseases, nutrition disorders, and regular medication consumption, except for standard vitamin D supplementation. The pubertal stage was assessed according to Tanner’s criteria. We recruited the maximum possible number of prepubertal children who consumed a lacto-ovo-vegetarian diet attending the Department of Nutrition at the Institute of Mother and Child in Warsaw between May 2018 and January 2020.

The control group included 50 healthy children (48.0% male, 52% female) aged 5–9 years on a traditional omnivorous diet, which consisted of consuming meat, poultry, and fish. Health status was assessed by collecting medical history data and conducting a basic physical examination.

The studied children (vegetarians and omnivores) were attaining the World Health Organization’s recommendation regarding physical activity. They accumulated about 60–90 min/day of MVPA (moderate-to-vigorous physical activity) and approximately 30 min/day of VPA (vigorous physical activity). VPA included activities after school twice a week for 1 or 2 h. The level of physical activity was similar in both of the studied groups of children.

The protocol of this study was in accordance with the Helsinki Declaration of Principles and approved by the Ethics Committee of the Institute of Mother and Child (decision number 12/2017, 12 March, 2017). All children’s parents were informed about the study procedures and all signed a written consent prior to the start of the study.

### 2.2. Methods

Measurements of body weight and height were performed, and body mass index (BMI) was calculated as body weight (kg) divided by height squared (m^2^). Body composition (fat mass, lean mass, bone mineral content—BMC) was measured by dual-energy X-ray absorptiometry (DXA) using Lunar Prodigy (General Electric Healthcare, Madison, WI, USA).

As described in more detail in a previous article [27], the dietary assessment was analyzed using the nutritional software program Dieta5^®^ (National Food and Nutrition Institute, Warsaw). The parents of the studied children were advised by a nutritionist and asked to prepare a food diary for their children. Three dietary recalls (two weekdays and one weekend day) were performed to evaluate the dietary habits. Average daily energy intake; percentage of energy from dietary protein, fat and carbohydrates, and fiber; and mineral and vitamin intakes were assessed in the studied children.

Blood samples were obtained from the children in the morning after an overnight fast. The serum samples were obtained and frozen at −20 °C until analysis. Concentrations of serum myokines and adipokines were determined by commercial enzyme-linked immunosorbent assay (ELISA), according to the manufacturer’s instructions. Serum irisin level was assayed using the kit from BioVendor (Brno, Czech Republic). The limit of detection in this method was 1 ng/mL; intra-assay and inter-assay coefficients of variation (CV) were 4.9–8.2% and 8.0–0.7%, respectively. Myostatin concentration was determined using the kit from SunRed Biotechnology (Shanghai, China) with a limit of detection of 5.11 ng/L, and intra-assay and inter-assay of precision less than 8% and 11%, respectively. Concentration of leptin was assessed using kit from DRG Instruments GmbH (Marburg, Germany) with an analytical sensitivity of 0.2 ng/mL; intra-assay CV ranged between 4.2 and 7.3% and inter-assay CV ranged between 3.7 and 9.1%. Level of total adiponectin was measured using a kit from TECOmedical AG (Sissach, Switzerland), with the lower detection limit of 0.6 µg/mL; intra-assay CV: 2.35–4.66%, inter-assay CV: 5.70–6.72%. Serum visfatin level was detected using the Nampt (Visfatin/PBEF) kit from Adipogen Life Science (Liestal, Switzerland). The lowest level of Nampt visfatin that can be detected by this assay was 30 pg/mL; the intra-assay precision was 2.31–9.11% and the inter-assay precision was 4.66–7.24%. Serum omentin was measured using the kit from SunRed Biotechnology (Shanghai, China), with a sensitivity of 5.22 ng/mL; intra-assay CV was less than 10%, and inter-assay CV was less than 12%.

### 2.3. Statistical Analyses

All analyses were carried out using IBM-SPSS software version 23.0 (SPSS INC., Chicago, IL, USA). The normality of the distribution of the variables was checked using the Kolmogorov–Smirnov test. To compare categorical data, the chi-squared test was used. Data were described as mean values and standard deviation (SD) for the variables with normal distribution, or as median values and interquartile ranges (1Q–3Q) for non-normally distributed variables. For group comparisons, the Student’s *t*-test was used when data were normally distributed, and the Mann–Whitney U test was used for non-parametric variables. Significance was accepted when the *p*-value was <0.05. Correlation analysis was performed using the Spearman test and interpreted with Bonferroni correction (threshold *p*-value of 0.002)**.**

A series of multivariate linear regression models were estimated separately for serum levels of myostatin and irisin. In the first step, an ordinary least square method of estimation was applied. All available data were considered as potential predictors; in total there were 23 variables representing: other myokines, adipokines, and anthropometric and nutritional parameters. To find an optimal set of predictors, a stepwise selection strategy was chosen. After each step, in which a new variable was added, all variables accepted previously were checked in terms of their significance. The cut-off points were less rigorous than usual (0.10 for entry and 0.15 for removal) to check the effect of predictors close to the significance level (more than 0.05 but less than 0.10). This facilitated finding more predictors with a significant contribution to the variability of dependent variables, despite relatively weaker significance. The results were presented as both standardized and unstandardized parameters. The change of R^2^ was considered an important goodness of fit statistic. In the second step, the same independent variables were included in the generalized linear models (GLM) with skewed link function γ, and *p*-values were compared. In GLM models, the maximum likelihood method of estimation is employed, and an other than normal distribution of the dependent variable is accepted. In addition, two GLM models, estimated on the combined sample, were added as Appendix A. They included dietary group as a factor and all variables accepted in the previous stage as covariates. The significance of the main effect and 2-way interactions between dietary groups and covariates were tested.

## 3. Results

Both groups of children were comparable in terms of age and anthropometric parameters (Table 1). In vegetarians, we noted significantly lower (*p* = 0.018) fat mass than in omnivorous peers.

Analyzing the children’s diets, we observed similar daily energy intake in both groups of children, whereas the proportions of macronutrient intakes in vegetarians and omnivores were different (Table 2). Vegetarians had a significantly higher percentage of energy from carbohydrates (*p* = 0.002), a lower percentage of energy from protein (*p* < 0.001), and a similar percentage of energy from fat (*p* = 0.166) compared with omnivores. Among minerals, vegetarians had a comparable intakes of phosphorus (P), magnesium (Mg), and calcium (Ca) but a significantly higher intake of manganese (Mn) (*p* = 0.020). In terms of vitamins, children on a vegetarian diet had a significantly lower intake of vitamin B_12_ (*p* < 0.001) and a higher intake of vitamin C (*p* = 0.019). No statistically significant differences in dietary intakes of vitamin A, D, and E in both groups were found.

As shown in Table 3, there are no statistically significant differences in the median values of serum myostatin and irisin concentrations or in the levels of adipokines such as visfatin, omentin, and adiponectin between vegetarians and omnivores. However, leptin concentration was significantly lower in vegetarian compared with omnivorous children.

Assessing simple correlations, we found significant associations between myokine and adipokine levels, including a strong positive association between serum myostatin and omentin levels in both groups of children (r = 0.415, *p* = 0.002 and r = 0.455, *p* = 0.002, respectively) (Table 4). Additionally, we observed, only in omnivores, a weak positive association between irisin and omentin concentrations (r = 0.300, *p* = 0.045) and, only in vegetarians, an inverse correlation between irisin and Nampt visfatin levels (r = −0.282, *p* = 0.037). We did not notice significant correlations between classical adipokines (leptin, adiponectin) and myokines levels. Concerning the anthropometric data, we noted significant positive correlations between myostatin level and body fat percentage in omnivores (r = 0.402, *p* = 0.004) and weak negative correlations of irisin levels with lean mass (r = −0.266, *p* = 0.049) and BMC (r = −0.269, *p* = 0.047) in vegetarians. No significant associations between myokine concentrations and other anthropometric parameters were found in either group of children.

Analyzing the associations between myokine levels and nutritional parameters, we observed that myostatin concentration was negatively correlated with the percentage of energy from protein (r = −0.338, *p* = 0.014), dietary calcium (r = −0.278, *p* = 0.046), and vitamin A (r = −0.304, *p* = 0.028) intakes in vegetarian children. In the omnivores, myostatin level was inversely correlated with dietary vitamin C (r = −0.306, *p* = 0.041) and vitamin E (r = −0.343, *p* = 0.021) intakes. No statistically significant correlations between irisin levels and nutritional data in either study group were found. After Bonferroni corrections, only the correlation between myostatin and omentin levels remained significant at *p* < 0.002, both in vegetarians and omnivores.

Multivariate analysis showed that four variables are independent predictors of the myostatin level in children on a vegetarian diet (Table 5). The final model explained 33.3% of the variability of this myokine according to the corrected R^2^ determination coefficient. We observed that serum myostatin was strongly positively related to omentin levels (*p* < 0.001) and dietary vitamin E intake (*p* = 0.003). We also noticed that two factors (dietary calcium and vitamin B_12_), which showed a weak relationship with the myostatin level in a simple correlation analysis, were statistically significant (*p* = 0.013 and *p* = 0.023, respectively) in this model.

Six independent variables, which explain almost half of the myostatin variability (48.6%), were qualified to the multifactor model in the omnivorous group. Serum myostatin was strongly, positively correlated to omentin levels (*p* = 0.001) and body fat percentage (*p* < 0.001) and weakly with dietary vitamin B_12_ (*p* = 0.013). Additionally, we showed that myostatin level was negatively related with the fat/lean ratio (*p* < 0.001), total energy intake (*p* = 0.011), and dietary vitamin D intake (*p* = 0.002). Attention should be paid to myostatin predictors, the effects of which were revealed only in multivariate analysis, with a weak correlation in simple correlation analyses (vitamin D, vitamin B_12_, and fat/lean ratio).

In the model estimated for the combined group (Appendix A in the Appendix A), the main effect of omentin, calcium, and vitamin E, and two 2-way interactions between the diet group and vitamin B_12_ and vitamin E, respectively, were significant. The relationship between theoretical myostatin based on regression models and dietary vitamin B_12_ in the two groups of children is shown in Figure 1.

Analogous models, in which the dependent variable was irisin, were characterized by a weaker adjustment level and contained less predictors (Table 6).

In the vegetarians, BMC and total energy intake explained 19.9% of the variability of irisin; however, in the omnivores, two factors, omentin and dietary manganese, explained 15.3% of the variability of irisin. Serum omentin level turned out to be the main predictor (*p* = 0.002). The final model also included dietary manganese intake, which explained 7.7% of the variability of irisin.

The introduction of two variables such as leptin and adiponectin into the models (Table 5 and Table 6) did not confirm that these factors independently affected the variability of myokines (myostatin and irisin).

The specification of the above four models was checked by a different estimation method, taking into account the skewness of the distribution of the two dependent variables. The inference concerning the significance of the previously identified predictors had not changed. Moreover, in relation to the association between dietary manganese intake and serum irisin level in the control group, the significance level of the regression parameter decreased from 0.071 to 0.013.

In the model estimated for the combined group (Appendix A in the Supplementary materials), the main effect of BMC and energy intake was significant, while the diet group and omentin remained insignificant. However, two significant 2-way interactions between the diet group by omentin and BMC (respectively) were found.

## 4. Discussion

The main finding of the present study was that prepubertal children consuming vegetarian and omnivorous diets had comparable serum concentrations of myokines (myostatin, irisin) and selected adipokines (visfatin, omentin). Similar to our previously report [28] concerning classical adipokines, the level of leptin was significantly lower in vegetarians than in omnivores; however adiponectin concentration was comparable between groups. There were also significant associations between myokine and adipokine levels, suggesting cross-talk between skeletal muscle and fat tissue. The strongest was a positive correlation between serum myostatin and omentin concentrations observed in simple correlation analysis and revealed in multivariate regression models in both of the studied groups of children. There were no significant correlations between myokines and leptin as well as adiponectin levels.

The interpretation of myokines and adipokine levels in regards to their clinical use in pediatrics is difficult because of the limited amount of research in this field and a relative lack of reference values of these parameters for children and adolescents. The literature contains the normal values for well-known adipokines, such as leptin and adiponectin, in healthy children, but little is known about the normal values of visfatin, omentin, or myokines in the pediatric population [29,30,31,32,33]. It is difficult to compare the serum myokine levels between studies, because the reported levels varied greatly due to the use of different methods and research carried out in different populations and age groups. Han et al. [34] observed the myostatin value to be 12.3 ± 3.6 ng/mL in healthy young males. Additionally, Lakshman et al. [35] reported that the serum concentration of myostatin was 8.0  ±  2.3 ng/mL in young men, with a tendency to be lower in older people. In a group of healthy adults, Yalcin et al. [36] found that the serum myostatin level was 7004.8 ± 677.5 pg/mL and the irisin level was 625.1 ± 128.6 ng/mL in healthy Turkish adults. Elizondo-Montemayor et al. [32] assessed irisin levels in small groups of Mexican children and found significantly lower irisin concentrations in the underweight group (164.3 ± 5.95 ng/mL) than in the normal weight group (185.29 ± 2.62 ng/mL) and the obese group (182.80 ± 5.58 ng/mL). Similarly to the current study, Colaianni et al. [33] observed a serum irisin value of 2.59 ± 1.15 ng/mL in a group of 34 healthy Italian children aged 9.8 ± 3.2 years.

Concerning Nampt visfatin, Nurten et al. [31] reported that median values of this adipokine were: 1.8 ng/mL in healthy children aged 9.6 ± 4.3 years and 1.4 ng/mL in children aged 13.2 ± 3.9 years. Additionally, median serum concentrations of omentin were 325.8 ng/mL and 290.9 ng/mL in the above groups of children. These values are similar to those in the current study, and the researchers used the same ELISA kits to determine these adipokines. It is important to establish the secretory profile of myokines and adipokines and to understand their role in the link between skeletal muscle, fat tissues, and other organs.

In our study, despite the lack of statistical differences between myokines and adipokines in the two studied groups, there was a trend towards an increase in values of myostatin and omentin by about 30%, Nampt visfatin by 20%, and irisin by 10% in vegetarians compared with omnivores. Analyzing the correlations, we showed that there were no significant relations between myostatin and the anthropometric parameters in both studied groups of children, except for a positive correlation (revealed in the multivariate regression model) between myostatin and body fat percentage in the omnivorous group. Additionally, irisin concentration did not correlate with body composition parameters, except for weak correlations with lean mass and bone mineral content in the vegetarian group. Our results are in line with other studies conducted in adults [37].

It is important to expand the knowledge of how tissues of great metabolic importance, such as skeletal muscle and adipose tissue, contribute synergistically to the maintenance of body homeostasis [8,38]. Myokines play a role in restoring a healthy cellular environment, reducing low-grade inflammation and thereby preventing metabolic related diseases. In view of that, the pleiotropic effects of myokines on multiple tissues lead to energy homeostasis. The muscle-adipose axis has an important effect in maintaining a balanced ratio of skeletal muscle to fat and thus may play a key role in the modulation of body composition [39]. Myokines and adipokines appear to be involved in autocrine/paracrine interactions within the muscles as well as adipose and other tissues. In this context, the strong positive association between myostatin and omentin levels observed in our vegetarians revealed cross-talk between skeletal muscle and fat tissues. We suggested that omentin, similar to adiponectin, has a metabolic protective effect on skeletal muscle, as it promotes glucose and fatty acid homeostasis [40,41]. The potential role of another adipokine, visfatin, in muscle metabolism has been discussed, but thus far final conclusions cannot be drawn.

Other metabolic mechanisms such as inflammation are involved in the development of muscle disorders. In our previous study, we demonstrated higher ratios of anti-inflammatory to pro-inflammatory adipokines in children consuming a vegetarian diet compared with omnivores [28]. In view of that, strong positive association between omentin (adipokine with anti-inflammatory properties) with myostatin levels could indicate a favorable anti-inflammatory status observed in our studied children.

Various factors, including nutrients, may influence muscle and adipose tissue. In the present study, myostatin significantly correlated with the percentage of energy from protein, dietary calcium, and vitamin A in vegetarians. Massive amounts of energy were utilized for muscle protein synthesis (MPS) and motor function under both resting and exercise conditions. Not only dietary protein quality but also quantity was an important factor impacting MPS for building and repairing muscle tissues [42]. Branched-chain amino acids, such as leucine, isoleucine, and valine, were particularly important. These amino acids were more concentrated in animal-based protein compared with plant proteins [43]. Additionally, the timing and distribution of protein consumption through the day and the digestion and absorption rates of different proteins can also alter and impact muscle protein synthesis.

Minerals and vitamins can promote the growth and repair of skeletal muscle. Nutrients from plant and animal sources have differing bioavailability. In our study, vegetarians as well as omnivores had lower calcium and vitamin D intake (calcium insufficiency in 80% of vegetarians and 65% of omnivores, vitamin D insufficiency in about 90% vegetarians and 80% omnivores) according to the recommendations [44]. It is widely known that vegetarians are at risk of vitamin B_12_ deficiency because this compound is mainly found in animal products [45,46]. In our study, about 25% of the children on a vegetarian diet had insufficient vitamin B_12_ intake, while all omnivores had adequate dietary intake of this vitamin.

Plant-based diets rich in fruits and vegetables are characterized by a higher intake of antioxidants that play an important role in protecting cells from reactive oxygen species damage. In the present study, we noted a higher dietary intake of vitamin C, the main water soluble antioxidant, in vegetarians. In addition, vegetarians had increased intakes of manganese, which is an essential constituent of SOD-1 (superoxide dismutase 1), the most important antioxidative enzyme. A higher dietary intake of nutrients with antioxidant properties in the vegetarian diet provides health benefits [47,48]. It is difficult to explain the individual dietary intake of nutrients because they act synergistically to maintain body homeostasis. The study of Gauze–Gnagne et al. [49], conducted on rats, assessed the influence of various high-fat diets rich in palm oil or olive oil on the modulation of myokine gene expression. The authors reported that high-fat diets (dependending on the type of fat used) can differentially modulate the expression of some myokines (decreased irisin and increased myonectin levels).

To the best of our knowledge, this is the first report on serum myokines in children consuming a vegetarian diet. This represents a first attempt to understand the link between the interaction of myokines and adipokines and the role of diet in the regulation of important metabolic functions; however, further investigations are needed to fully understand the mechanisms.

Some limitations should be mentioned. Firstly, our conclusion was established based on a limited number of samples and needs to be further confirmed in a larger sample size. However, our studied groups of children were comparable in terms of age, Tanner stage, and body composition. Secondly, we did not determine other novel myokines (e.g., myonectin) but we are planning assessments of a wide range of myokines, such as interleukins, myonectin, and decorin in our future study conducted on children following different dietary styles. Thirdly, our results were based on single measurements of serum myokine and adipokine concentrations in children and therefore may not reflect the long-term exposure of these proteins. Additionally, we analyzed only serum levels of myokines and did not assess its expression by the skeletal muscles. However, we provided unique determination of myokines in prepubertal children with different types of diet for the first time. Fourthly, we did not provide an exact analysis of the physical activity of the studied children, but both groups were comparable regarding MVPA and VPA. We assessed the percentage of energy from protein and did not analyze the dietary amino acid intake in the studied groups of children; however, we are planning such assessments in our next study. Finally, the cross-sectional nature of our study does not permit causality statements. However, we believe that our results may create basic knowledge for further studies evaluating the possible role of diet on the myokine and adipokine profiles, inflammation, and metabolic status.

To conclude, consuming a vegetarian diet, particularly its lacto-ovo-vegetarian version did not influence the serum levels of myokines and adipokines such as adiponectin, visfatin, and omentin in the prepubertal period. However, leptin level was significantly lower in vegetarians compared with omnivores. The observed significant positive correlations between serum myostatin and omentin levels may represent a link between muscle and fat tissues. In view of this, omentin is an adipokine with anti-inflammatory properties, and its strong positive association with myostatin could indicate a favorable anti-inflammatory status. Further investigations could be important to fully understand the myokine/adipokine-related mechanisms involved in the cross-talk between skeletal muscle, adipose tissues, and nutrition.

## Figures and Tables

**Figure 1 jcm-10-03995-f001:**
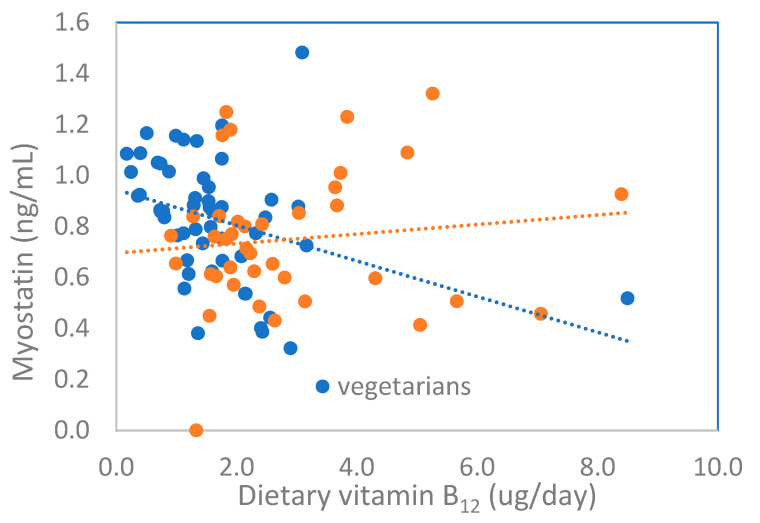
Scatter plot of theoretical myostatin estimated from regression model (Table 5) versus dietary vitamin B_12_ in vegetarian and omnivorous children.

**Table 1 jcm-10-03995-t001:** Anthropometric parameters in children on vegetarian and omnivorous diets.

Variables	Vegetarians (*n* = 55)	Omnivores (*n* = 50)	*p*
Age (years) ^b^	5.5 (5.0–7.0)	6.0 (5.0–8.0)	0.147
Body weight (kg) ^a^	20.70 ± 3.87	21.22 ± 5.86	0.723
Height (cm) ^a^	117.39 ± 10.42	118.58 ± 10.89	0.544
BMI (kg/m^2^) ^a^	14.76 ± 0.92	14.66 ± 1.44	0.070
Fat mass (kg) ^b^	3.32 (2.67–4.35)	3.94 (3.10–5.30)	0.018
Fat (%) ^b^	18.80 (14.80–22.40)	21.55 (17.15–24.23)	0.057
Lean mass (kg) ^b^	14.80 (13.55–16.88)	15.81 (12.77–19.52)	0.349
Fat/lean mass ^b^	0.233 (0.173–0.301)	0.272 (0.200–0.321)	0.114
BMC (g) ^a^	668.7 ± 177.6	697.0 ± 237.8	0.103

Data are presented as ^a^ mean values and standard deviation (SD) and ^b^ median values and interquartile ranges (1Q–3Q); BMI—body mass index, BMC—bone mineral content.

**Table 2 jcm-10-03995-t002:** Dietary intake of macro- and micronutrients in children on vegetarian and omnivorous diets.

Variables	Vegetarians (*n* = 55)	Omnivores (*n* = 50)	*p*
Total energy (kcal) ^a^	1404.6 ± 393.1	1590.5 ± 500.9	0.470
Percentage of energy from protein ^b^	11.5 (10.3–13.1)	13.6 (11.6–15.0)	<0.001
Percentage of energy from fat ^b^	31.1 (27.6–34.6)	33.3 (28.1–37.1)	0.166
Percentage of energy from carbohydrates ^b^	56.7 (53.3–61.3)	52.9 (49.3–58.0)	0.002
Fiber (g/day) ^a^	19.0 ± 8.3	16.0 ± 6.1	0.063
Calcium (mg/day) ^b^	513.3 (311.9–641.3)	604.0 (404.9–778.0)	0.056
Phosphorus (mg/day) ^b^	841.2 (597.7–1011.4)	868.5 (707.9–1017.2)	0.161
Magnesium (mg/day) ^b^	223.5 (174.2–320.2)	216.7 (170.2–260.3)	0.633
Manganese (mg/day) ^b^	3.49 (2.33–4.37)	2.54 (1.89–3.52)	0.020
Vitamin A (µg/day) ^b^	897.6 (658.8–1447.6)	837.8 (589.2–1298.1)	0.203
Vitamin B_12_ (µg/day) ^b^	1.44 (0.90–2.12)	2.31 (1.80–3.78)	<0.001
Vitamin C (mg/day) ^b^	79.8 (44.8–122.3)	68.7 (47.8–103.3)	0.019
Vitamin E (mg/day) ^b^	9.50 (6.94–11.81)	8.04 (6.03–11.68)	0.315
Vitamin D (µg/day) ^b^	1.73 (0.82–4.15)	2.12 (1.01–5.63)	0.058

Data are presented as ^a^ mean values and standard deviation (SD) and ^b^ median values and interquartile ranges (1Q–3Q).

**Table 3 jcm-10-03995-t003:** Serum concentrations of myokines and adipokines in vegetarian and omnivorous children.

Variables	Vegetarians	Omnivores	*p*
Myostatin (ng/mL)	0.82 (0.52–1.19)	0.58 (0.47–1.01)	0.131
Irisin (µg/mL)	3.05 (2.00–4.40)	2.79 (2.24–4.02)	0.957
Nampt visfatin (ng/mL)	1.82 (0.77–2.85)	1.44 (0.89–3.056)	0.969
Omentin (ng/mL)	504.7 (295.3–1156.9)	386.2 (263.9–1010.8)	0.149
Leptin (ng/mL)	1.39 (0.80–1.90)	2.05 (1.22–2.80)	0.003
Adiponectin (µg/mL)	8.76 (7.27–11.60)	7.95 (6.61–10.03)	0.059

Data are presented as median values and interquartile ranges (1Q–3Q).

**Table 4 jcm-10-03995-t004:** Spearman’s correlations of myokine concentrations with anthropometric parameters, nutritional parameters, and adipokine levels in vegetarian and omnivorous children.

Variables	Vegetarians				Omnivores			
	Myostatin		Irisin		Myostatin		Irisin	
	r	*p*	r	*p*	r	*p*	r	*p*
Anthropometric parameters:								
BMI	0.116	0.397	0.125	0.364	0.021	0.888	−0.088	0.544
Fat mass	−0.002	0.989	0.081	0.555	0.105	0.466	0.038	0.791
Fat (%)	0.070	0.609	0.103	0.452	0.402	0.004	−0.077	0.594
Lean mass	−0.019	0.890	−0.266	0.049	0.015	0.917	0.221	0.122
Fat/lean mass ratio	0.020	0.882	0.133	0.334	0.090	0.536	−0.113	0.435
BMC	−0.146	0.289	−0.269	0.047	−0.009	0.952	0.073	0.615
Nutritional parameters:								
Total energy intake	0.029	0.840	0.185	0.189	−0.048	0.752	−0.100	0.515
Percentage of energy from protein	−0.338	0.014	−0.128	0.366	0.113	0.460	−0.118	0.441
Percentage of energy from fat	0.083	0.559	−0.208	0.140	−0.294	0.050	0.017	0.914
Percentage of energy from carbohydrates	0.045	0.753	0.254	0.070	0.243	0.108	0.086	0.574
Dietary fiber	−0.015	0.919	0.042	0.770	−0.063	0.681	0.037	0.810
Dietary calcium	−0.278	0.046	0.105	0.457	0.128	0.403	−0.016	0.919
Dietary phosphorus	−0.100	0.480	0.169	0.231	0.067	0.660	−0.159	0.296
Dietary magnesium	0.021	0.883	0.187	0.184	0.011	0.945	−0.231	0.127
Dietary manganese	0.101	0.475	0.005	0.972	0.030	0.844	−0.160	0.295
Dietary vitamin A	−0.304	0.028	−0.013	0.925	−0.040	0.759	0.233	0.129
Dietary vitamin B_12_	−0.207	0.140	0.049	0.731	0.085	0.579	−0.070	0.649
Dietary vitamin C	0.034	0.810	0.192	0.173	−0.306	0.041	0.044	0.776
Dietary vitamin D	−0.161	0.254	0.044	0.755	−0.149	0.329	0.102	0.505
Dietary vitamin E	0.155	0.272	−0.006	0.965	−0.343	0.021	−0.089	0.562
Adipokines:								
Omentin	0.415	0.002	−0.072	0.603	0.455	0.002	0.300	0.045
Nampt visfatin	−0.015	0.912	−0.282	0.037	−0.094	0.515	−0.214	0.135
Leptin	0.053	0.701	−0.036	0.796	0.162	0.618	0.114	0.431
Adiponectin	0.005	0.973	−0.058	0.675	0.072	0.618	0.102	0.479

BMI—body mass index, BMC—bone mineral content.

**Table 5 jcm-10-03995-t005:** Multivariate regressions of myostatin with independent variables in the groups of vegetarians and omnivores.

Independent Variables	Regression Parameters			t	p1 p2	ΔR^2^
	Unstandardized		Standardized			
	B	SE	β			
Vegetarians						
Omentin	0.0003	0.0001	0.437	3.746	0.000 0.000	0.130
Dietary calcium	−0.0005	0.0002	−0.301	−2.234	0.030 0.013	0.128
Dietary vitamin E	0.0293	0.0097	0.406	3.012	0.004 0.003	0.066
Dietary vitamin B_12_	−0.1024	0.0474	−0.327	−2.159	0.036 0.023	0.061
R^2^ crude = 0.385; R^2^ adjusted = 0.333						
Omnivores						
Omentin	0.0003	0.0001	0.359	3.038	0.005 0.001	0.115
Fat (%)	0.0612	0.0118	0.958	5.168	0.000 0.000	0.163
Fat/lean	−2.4221	0.7434	−0.588	−3.258	0.003 0.000	0.090
Total energy intake	−0.0002	0.0001	−0.255	−2.159	0.038 0.011	0.063
Dietary vitamin D	−0.0327	0.0117	−0.350	−2.806	0.008 0.002	0.076
Dietary vitamin B_12_	0.0495	0.0218	0.302	2.273	0.030 0.013	0.060
R^2^ crude = 0.567; R^2^ adjusted = 0.486						

B—unstandardized regression coefficient; SE—standard error; β—standardized regression coefficient; p1—significance in ordinary linear regression model; p2—significance in generalized linear model (GLM); ΔR^2^—change in R^2^; constant was included.

**Table 6 jcm-10-03995-t006:** Multivariate regressions of irisin with independent variables in the groups of vegetarians and omnivores.

Independent Variables	Regression Parameters			t	p1 p2	ΔR^2^
	Unstandardized		Standardized			
	B	SE	β			
Vegetarians						
BMC	−0.0010	0.0003	−0.469	−3.616	0.001 0.000	0.158
Total energy intake	0.0002	0.0001	0.279	2.152	0.036 0.043	0.073
R^2^ crude = 0.230; R^2^ adjusted = 0.199						
Omnivores						
Omentin	0.0004	0.0001	0.406	2.663	0.012 0.002	0.121
Dietary manganese	−0.0839	0.0450	−0.284	−1.863	0.071 0.013	0.077
R^2^ = 0.198; R^2^ adjusted = 0.153						

B—unstandardized regression coefficient; SE—standard error; β—standardized regression coefficient; p1—significance in ordinary linear regression model; p2—significance in generalized linear model (GLM); ΔR^2^—change in R^2^; constant was included.

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
