# Peer review of "Does a Vegetarian Diet Affect the Levels of Myokine and Adipokine in Prepubertal Children?"

_jcm, 2021, doi:10.3390/jcm10173995_

Round 1

Reviewer 1 Report

As compared to nonrestrictive omnivorism, vegetarianism has been practiced by the human in many cultures for centuries. While the vegetarian diet has gained substantial popularity recently, particularly in western society, the health benefit of the effects on the metabolism and physiology of individuals remains largely debated. In this study, the authors Ambroszkiewicz et al. assessed the levels of four myokines and adipokines among children under a vegetarian or omnivorous diet. While the authors did not find significant effects of a vegetarian diet on myokines and adipokines levels compared to the ominous control diet, the authors found significant correlations of the circulating factors with some dietary and anthropometric parameters.

Overall, experimental design, data analysis, and result interpretations are appropriate for the study for the most part. However, enthusiasm is slightly damped due to the following concerns:

Major concerns:

There are hundreds of muscle or adipose tissue-derived circulating factors. While the authors described some background of visfatin and omentin, the rationale for focusing the assessment on these two adipokines is lacking. While the authors aimed to assess the tissue crosstalk between the skeletal muscle, evidence for the direct involvement of the assessed factors in muscle-adipose tissue crosstalk is lacking. On the other hand, myokines such as myonectin (FAM132B), which has been shown to be skeletal muscle-derived and maintain systemic homeostasis via adipose tissue lipid metabolism, have not been included in the assessment.

Despite the reported significant differences in energy content, energy substrate proportion, micronutrient abundance, adiposity in individuals on a vegetarian diet, the levels of the assessed myokines and adipokines are comparable. Therefore, it seems to suggest that the selected circulating factors of interest do not represent the state of metabolic homeostasis. Myostatin positively associates with Omentin levels in both vegetarian and omnivorous groups. Circulating levels of both factors are highly variable. This variability is particularly true for omentin, as evidenced by the reported large interquartile ranges. Unfortunately, it is challenging to determine the relationship between myostatin and omentin without performing an analysis of covariance (ANCOVA). Therefore, it is still unclear whether a vegetarian diet alters the levels of circulating factors when some circulating factors (e.g., omentin) are covariate for the other.

Similarly, it has been previously shown and demonstrated in the manuscript that a vegetarian diet reduces adiposity. Interestingly, this reduction seems to abolish the association of adiposity and myostatin levels. Unfortunately, with the way of data presentation, it is challenging to estimate the effect of the diet on myostatin levels without performing ANCOVA, using adiposity as the covariates.

In short, here are the two practical pieces of advice to substantially improve the quality of the manuscript in the revision:

  1. Measure the myonectin (FAM132B/ERFE) levels and provide detailed rationales for all circulating factors assessed. Human anti-myonectin ELISA kits are commercially available.
  2. Perform ANCOVA to compare the circulating factors in vegetarian and omnivorous groups, using adiposity or other measured parameters as the covariates. The ANCOVA can be essentially achieved by comparing the linear regression lines of the two groups, which is one use of ANCOVA.

Minor concerns:

  1. Please correct the decimals of values for children’s heights in table 1.
  2. Please avoid referring to micronutrients as a source of energy in the abstract (page 1, line 24)

Author Response

Reviewer 1:

Thank you very much for your valuable comments and suggestions which substantially improved the quality of this manuscript.

1. We agreed with the Reviewer`s suggestion that measure levels of myonectin would be valuable. We know that human anti-myonectin ELISA kits are available. In the conducted studies we did not take into account the determination of myonectin, among other reasons because, as the Reviewer rightly noted its activity on systemic homeostasis through the influence on the adipose tissue metabolism is documented in the literature. We wanted to determine whether other myokines also had similar effects, and the first of them were myostatin and irisin assessed in the present project. We received the consent of the Bioethics Committee for the determination of these myokines and adipokines (visfatin and omentin). However, we are planning assessments of other myokines, such as interleukins, myonectin, decorin in our next study conducted on children following different dietary styles.

2. The association of myostatin and irisin with other covariates was analysed using multivariate regression methods such as the GLM, which corresponds to ANCOVA analysis. It means that the association with omentin was adjusted for other factors if only they were significant. In the methods section we explained that all variables presented in Tables 1 and 2 were taken into account. However, the regression results presented in Tables 5 and 6 were limited to the significant predictors of myostatin and irisin, respectively. The final models were different in the vegetarian and omnivorous groups. However, we found the reviewer's comment inspiring and estimated additional models for the combined sample of the two groups, introducing dietary factors into the model. These models are included as supplementary electronic material. We investigated the so-called main effects and 2-way interactions, where the diet group is one part. The interaction with vitamin B12 (seen earlier in Table 5) was the most interesting. Therefore, we added one scatter plot figure where the dependent variable (myostatin) is theoretically estimated from the regression (Table 5). The description of the methods has been supplemented by these additional analyses of interactions.

We added new statistical analyses presented in Figure 1 (in the manuscript) and Table A1 and Table A2 (supplementary electronic material).

Minor concerns:

1. Please correct the decimals of values for children’s heights in table 1.

 We corrected this error.

2. Please avoid referring to micronutrients as a source of energy in the abstract (page 1, line 24)

 We changed this sentence.

Additionally, we have corrected spaces, typos, and grammatical language errors throughout the manuscript (in red).

Reviewer 2 Report

In the manuscript “Does a vegetarian diet affect the levels of myokine and adi-2 pokine in prepubertal children?” by Dr. Ambroszkiewicz and collaborators show the levels of different adipokines and myokines in prepubertal boys and girls in relation to their vegetarian and omnivorous diets.

The manuscript is very interesting, well structured, and interest in the field.

However, there are some points that need to be clarified.

Major issues:

1.-It is necessary to define to origin of volunteers in terms of race (all of them are Caucasians?)

2.-Apart from the approval of the Ethics Committee, this type of studies is registered in the Clinical Trials or EudraCT. It is the case? Please, provide your registration number.

3.-Serum samples were stored at -20ºC, but it is advisable to store at lower temperatures. However, authors have been collecting all the samples from May 2018 to January 2020, which can introduce an important bias in the obtained results. Please, explain this point.

4.-It is important to analyse the relationship between the classic clinical biomarkers (i.e. blood glucose, HDL, LDL, …) with the myokines and adipokines analyzed. Perhaps these data were not available in the extraction data, but possibly these data were in the medical history.

5.-To have a more complete view of the impact of vegetarian diet in children in the myokine and adipokine levels, it is important to provide the levels of adiponectin and leptin.

6.-It is necessary to revise the levels of Myostatin, because they are below 1 ng/L, while the detection limit provided in the ELISA kit is 5.11 ng/L. Can you explain this inconsistency?

7.-It is necessary to use a Bonferroni test or a similar test to adjust the p-values of Spearman correlations.

8.-Table 2 shows the dietary intake. However, considering the wide age range, they should be relativized by age or weight.

Minor points:

-There are some typing mistakes throughout the manuscript (i.e. an additional space on line 14, between the words “following” and “vegetarian”, or “thr” in line 104; or bold inside tables -Table 5 vegetarians vs. Omnivores). Please, revise throughout the manuscript.

-To facilitate reading, indicate in bold the statistically significant correlations in Table 4.

Author Response

Reviewer 2:

Thank you very much for your valuable comments and suggestions which substantially improved the quality of this manuscript. We corrected all issues point by point.

Major issues:

 1.-It is necessary to define to origin of volunteers in terms of race (all of them are Caucasians?)

We added the information that all studied children were Caucasians in the Materials and Methods section.

2.-Apart from the approval of the Ethics Committee, this type of studies is registered in the Clinical Trials or EudraCT. It is the case? Please, provide your registration number.

This study was conducted as a statutory scientific activity of the Institute of Mother and Child (OPK 510-13-22) and did not require registration in databases such as the Clinical Trials or other. However, we obtained the approval of the local Ethics Committee (decision number 12/2017).

3.-Serum samples were stored at -20ºC, but it is advisable to store at lower temperatures. However, authors have been collecting all the samples from May 2018 to January 2020, which can introduce an important bias in the obtained results. Please, explain this point.

 Indeed, the serum samples for this study were collected between 2018-2020, but the samples were stored at -20ºC for no longer than 1-2 months until the determination of myokines and adipokines. The serum samples were stored in separate portions for each parameter determination and were not thawed twice. The adipokines and myokines were successively assessed by ELISA methods in the collected samples along with other samples collected for other projects (e.g. children with cow's milk allergy or children with Prader-Willi syndrome).  

4.-It is important to analyse the relationship between the classic clinical biomarkers (i.e. blood glucose, HDL, LDL, …) with the myokines and adipokines analyzed. Perhaps these data were not available in the extraction data, but possibly these data were in the medical history.

There are several papers in the literature on classical biochemical markers in vegetarians, including children. In our study conducted in 2011 (Ambroszkiewicz et al. Med Wieku Rozwoj. 2011 Jul-Sep;15(3):326-34) we examined serum concentrations of cholesterol (HDL, LDL), TG, leptin,  leptin soluble receptor, adiponectin in children on a vegetarian diet.

Thus. in this project submitted by us to the Bioethics Committee, in the methodology, we did not include the determination of classic clinical biomarkers (their determination would require consent to collect more blood, and the study groups were children who had other routine tests at the same time. We measured myokines (myostatin and irisin) and adipokines (visfatin and omentin) in groups of vegetarian and omnivorous children, and for determination of these parameters we received the consent of the Bioethics Committee.

5.-To have a more complete view of the impact of vegetarian diet in children in the myokine and adipokine levels, it is important to provide the levels of adiponectin and leptin.

 Leptin and adiponectin are quite well-known adipokines. We analyzed their levels in earlier study (Ambroszkiewicz et al. Nutrients. 2018 Sep 6;10(9):1241. doi: 10.3390/nu10091241) and observed significantly lower levels of leptin in vegetarians compared with omnivores and similar level of total adiponectin in both studied groups of children. For this project, we chose lesser-known adipokines that had not yet been described in children on a vegetarian diet. In addition, we obtained the consent of the Bioethics Committee for the determination of a limited number of parameters, which was associated with the possibility of collecting a specific amount of blood for testing.

 6.-It is necessary to revise the levels of Myostatin, because they are below 1 ng/L, while the detection limit provided in the ELISA kit is 5.11 ng/L. Can you explain this inconsistency?

Thank you for this suggestion. There was an error in the table concerning units. The results are converted and given in ng/mL (not in ng/L). We corrected it in the table and in the results.

7.-It is necessary to use a Bonferroni test or a similar test to adjust the p-values of Spearman correlations.

Information on Bonferroni correction was included in the description of methods and results.

8.-Table 2 shows the dietary intake. However, considering the wide age range, they should be relativized by age or weight.

The nutritional standards for the Polish population revised in 2020 include, among others, standards for energy, nutrients, micronutrients, and vitamins for individual age groups of children. The examined children, according to the division criteria used in the standards, belonged to two age groups, i.e. 4-6 and 7-9 years. Our analysis showed that the percentage of children aged 4-6 and 7-9 on a vegetarian diet was 69.1% and 30.9%, respectively, and 68.0% and 32.0% on a traditional diet. At the same time, we did not find significant differences in the values of BMI of children in age subgroups and the percentage of girls and boys in subgroups was similar. Therefore, in the conducted analyzes, we treated both groups collectively with a view to obtaining the appropriate sample size.

 Minor points:

-There are some typing mistakes throughout the manuscript (i.e. an additional space on line 14, between the words “following” and “vegetarian”, or “thr” in line 104; or bold inside tables -Table 5 vegetarians vs. Omnivores). Please, revise throughout the manuscript.

-To facilitate reading, indicate in bold the statistically significant correlations in Table 4.

We corrected all above suggested minor points. Additionally, we have corrected spaces, typos, and grammatical language errors throughout the manuscript (in red).

Round 2

Reviewer 1 Report

It is agreeable that the authors has made some effort in revising the manuscript. However, my previous major concerns have not been adequately addressed. Due to these major concerns, unfortunately, the rationale for the assessment of the specific circulating factors and the mechanistic insights from study are still, for the most part, underdeveloped in this manuscript. Therefore, I remain unconvinced that the study as it currently stands substantially improved our understanding of the vegetarian diet on muscle-adipose tissue crosstalk in children beyond an incremental advancement.

Author Response

Dear Reviewer,

Thank you for your suggestions.

We have reviewed our responses to the first round of your suggestions. Most of the comments have been taken into account and relevant changes have been marked in the text. We enriched the presentation of results with one figure (in the main text) and two tables (A1 and A2 in the appendix) and expanded the Discussion.  

The comments regarding the novel myokines (e. g. myonectin) that should be analysed are most justified. However, we did not have such a wide range of laboratory experiments included in our project due to limited funds and we lack of consent of the Bioethics Committee for these studies for this moment. We added this information in the limitation section.

Reviewer 2 Report

Nothing to add. Excellent work. 

Author Response

Thank you very much for your valuable comments which substantially improved the quality of our manuscript.